

# Substrate rugosity and temperature matters: patterns of benthic diversity at tropical intertidal reefs in the SW Atlantic

Ana Carolina de A. Mazzuco, Patricia Sarcinelli Stelzer and Angelo F. Bernardino

Department of Oceanography, Universidade Federal do Espírito Santo, Vitória, Espírito Santo, Brazil

Corresponding author
Ana Carolina de A. Mazzuco,
ana.mazzuco@ufes.br,
ac.mazzuco@me.com

## ABSTRACT

Modeling and forecasting ocean ecosystems in a changing world will require advances in observational efforts to monitor marine biodiversity. One of the observational challenges in coastal reef ecosystems is to quantify benthic and climate interactions which are key to community dynamics across habitats. Habitat complexity (i.e., substrate rugosity) on intertidal reefs can be an important variable explaining benthic diversity and taxa composition, but the association between substrate and seasonal variability is poorly understood on lateritic reefs in the South Atlantic. We asked if benthic assemblages on intertidal reefs with distinct substrate rugosity would follow similar seasonal patterns of succession following meteo-oceanographic variability in a tropical coastal area of Brazil. We combined an innovative 3D imaging for measuring substrate rugosity with satellite monitoring to monitor spatio-temporal patterns of benthic assemblages. The dataset included monthly in situ surveys of substrate cover and taxon diversity and richness, temporal variability in meteo-oceanographic conditions, and reef structural complexity from four sites on the Eastern Marine Ecoregion of Brazil. Additionally, correlation coefficients between temperature and both benthic diversity and community composition from one year of monitoring were used to project biodiversity trends under future warming scenarios. Our results revealed that benthic diversity and composition on intertidal reefs are strongly regulated by surface rugosity and sea surface temperatures, which control the dominance of macroalgae or corals. Intertidal reef biodiversity was positively correlated with reef rugosity which supports previous assertions of higher regional intertidal diversity on lateritic reefs that offer increased substrate complexity. Predicted warming temperatures in the Eastern Marine Ecoregion of Brazil will likely lead to a dominance of macroalgae taxa over the lateritic reefs and lower overall benthic diversity. Our findings indicate that rugosity is not only a useful tool for biodiversity mapping in reef intertidal ecosystems but also that spatial differences in rugosity would lead to very distinct biogeographic and temporal patterns. This study offers a unique baseline of benthic biodiversity on coastal marine habitats that is complementary to worldwide efforts to improve monitoring and management of coastal reefs.

## INTRODUCTION

Marine benthic ecosystems hold a significant portion of global biodiversity that is mostly concentrated in threatened coastal habitats such as intertidal reefs (*Halpern et al., 2008*; *Andrades et al., 2018*). There are high concerns of human impacts on marine biodiversity across a range of expected climate change scenarios (*Dulvy, Sadovy & Reynolds, 2003*; *Solan et al., 2004*; *Harnik et al., 2012*; *McCauley et al., 2018*). In the tropics, forecasts point towards extreme climate-driven environmental conditions and severe biodiversity losses in the near decades (*Cheung et al., 2009*; *Bellard et al., 2012*; *Bathiany et al., 2018*). Understanding how patterns of biodiversity respond to increasing temperatures and subsequent oceanic-climatic changes is an important step towards effective management and conservation of marine ecosystems (*Santamaría & Méndez, 2012*; *Muelbert et al., 2019*).

In benthic marine communities, substrate and oceanographic parameters are critical predictors of the distribution and abundance of species, setting biodiversity patterns at multiple spatio-temporal scales and shaping observed biodiversity patterns (*Menge et al., 1997*; *Wieters, Broitman & Branch, 2009*; *Burrows, Harvey & Robb, 2008*; *Blanchette et al., 2008*; *Griffiths et al., 2017*). The strength of regulating factors at each location is linked to ecosystem dynamics (*Navarrete et al., 2005*), incorporating both local (e.g., substrate, productivity) and large-scale climatic and oceanographic conditions (e.g., *Steneck & Dethier, 1994*; *Menge et al., 2003*; *Zawada, Piniak & Hearn, 2010*; *Piacenza et al., 2015*; *Mazzuco et al., 2019*). On rocky shores, wave exposure and changes in sea temperature are major factors structuring and regulating benthic communities at multiple scales (e.g., *Burrows, Harvey & Robb, 2008*; *Blanchette et al., 2008*). In the intertidal zone, the reef physical structure and complex topography significantly influence the distribution, zonation and overall benthic biodiversity (e.g., *Archambault & Bourget, 1996*; *Ferreira, Gonçalves & Coutinho, 2001*; *Bloch & Klingbeil, 2016*). Therefore, coupling benthic and pelagic approaches will likely improve the ecological modeling of benthic communities and increase the success of modeling spatio-temporal variations and losses in biodiversity under a changing climate (*Griffiths et al., 2017*).

Temperature is a major driver of intertidal benthic biodiversity (e.g., *Thompson, Crowe & Hawkins, 2002*; *Hiscock et al., 2004*; *Morelissen & Harley, 2007*; *Harley, 2011*; *Meager, Schalacher & Green, 2011*; *Kordas et al., 2015*). Air and sea temperatures regulate spatio-temporal patterns of the whole in the intertidal zone by creating a gradient of environmental conditions (*Sunday et al., 2019*; *Wallingford & Sorte, 2019*). Higher temperatures reduce intertidal biodiversity, particularly when temperatures overcome species' physiological limits (e.g., *Kelmo et al., 2014*; *Wethey et al., 2011*; *Vafeiadou et al., 2018*; *Starko et al., 2019*). In coral reef communities, increased temperatures negatively affect intertidal biodiversity due to coral mortality (*Anthony & Kerswell, 2007*; *Smit & Glassom, 2017*). While in temperate and upwelling rocky shores, warmer temperatures may have positive influences on overall biodiversity (*Valdivia et al., 2013*; *Puente et al., 2017*; *Lamy et al., 2018*). However, the relationship between temperature fluctuations and biodiversity patterns can be variable among marine ecosystems (e.g., *Olabarria et al., 2013*; *Kordas et*

*al., 2015*; *Meadows et al., 2015*; *Puente et al., 2017*; *Sorte et al., 2017*), and as a result, greatly change predictions of how climate change scenarios will impact global coastal biodiversity.

Rugosity is likewise a key-variable for intertidal biodiversity as it creates a range of micro-habitats that promotes species coexistence, so increased rugosity is expected to have a positive impact on the diversity of intertidal communities (e.g., *Londoño Cruz et al., 2014*; *Dias et al., 2018*; *Leclerc, 2018*; *Price et al., 2019*). Habitat complexity explains spatial changes in species distribution and assemblage composition along coastal regions (*Guichard, Bourget & Robert, 2001*; *Cruz-Motta et al., 2010*; *Zawada, Piniak & Hearn, 2010*; *Bloch & Klingbeil, 2016*) and can overcome other seascape variables in the regulation of biodiversity (*Fuchs, 2013*; *Meißner et al., 2014*; *Williams et al., 2015*). However, reef rugosity is not often used as a metric for coastal monitoring and less is known about how benthic assemblages at variable levels of reef rugosity respond to seasonal meteo-oceanographic variations (*Meager & Schlacher, 2013*). Understanding variability within and between habitat types is crucial to improve ecological models and impact management applications (*Huntington et al., 2010*; *McArthur et al., 2010*; *Kovalenco, Thomaz & Warfe, 2012*).

Rocky shores are ubiquitous along the Brazilian Atlantic coast, but they vary widely in their substrate structure and are also exposed to a range of climatic conditions. In the Eastern Brazil Marine Ecoregion, decadal temperatures exhibit significant warming trends that are likely to impact coastal marine biodiversity (*Bernardino et al., 2015*). Rocky lateritic (or sandstone) rocks are ubiquitous in the Eastern Brazil Marine Ecoregion and further north, whereas granite rocky coasts predominate in the southern sub-tropical coast. Reefs of lateritic sandstones are highly topographically complex and understudied globally when compared to granite reefs (*Amaral & Jablonski, 2005*; *Coutinho et al., 2016*). Lateritic reefs are potentially transitional habitats with typical rocky shore features but with highly eroded substrate (sandstone) interspaced by calcareous formations (*Albino, Neto & Oliveira, 2016*). Given a distinct substrate rugosity, lateritic reefs hold a singular and diverse benthic community dominated by macroalgal beds with several calcareous habit-forming species (*O'Hara, 2001*; *Guidetti et al., 2004*; *Azzarello et al., 2014*). Although community structure varies among reefs and throughout the year, the environmental mechanisms associated with such variability remain uncertain in this marine ecoregion of Brazil.

Unlike temperate and equatorial rocky shores, where increases in sea temperatures are causing biodiversity losses (*Smith, Fong & Ambrose, 2006*; *Hawkins et al., 2009*; *Jueterbock et al., 2013*), lateritic reefs may experience an increase in overall biodiversity during warmer periods since these reefs host a number of both tropical and subtropical species. In subtidal tropical reefs, warmer temperatures have been suggested to lead to an overall increase in taxa richness when compared to equatorial or subtropical regions (*Aued et al., 2018*). If temporal trends of diversity with observed warming at the coastal intertidal reefs are confirmed, these ecosystems will likely experience significant changes in the dominance of benthic taxa with marked functional changes (*Poloczanska et al., 2016*). For example, a decrease in macroalgae taxa and increase in anthozoans with warmer temperatures would not only change the current dominant habitat-forming species but also would impact food provisioning for a number of coastal benthic and pelagic organisms (*Andrades et al., 2019*; *Mazzuco et al., 2019*). These transitional intertidal communities may

respond to temperature with varying levels of relisience depending on their structure and dynamics (*Bernhardt & Leslie, 2013*; *Timpane-Padgham, Beechie & Klinger, 2017*; *Palumbi et al., 2019*).

Accessing and predicting changes in biodiversity from local to global scales is a high research priority. Several ocean observatories are incorporating biodiversity among their monitored variables to meet multiple-stakeholder needs (*Bax et al., 2019*; *Muller-Karger et al., 2017*). Standard protocols and technology to improve and speed biodiversity data collection are highlighted as potential solutions to monitor biodiversity at large scales and with high temporal resolution (*Canonico et al., 2019*). Associating abiotic surrogates and marine biodiversity change is an important tool to support forecasts in global change scenarios (*Canonico et al., 2019*). Satellite remote sensing products are adequate for biodiversity synoptic approaches, providing a reliable framework for different coastal regions (*Capotondi et al., 2019*). Finding current drivers of biodiversity change across marine habitats is fundamental to give us a better understanding of the expected patterns of biodiversity in the present and future.

Considering the importance of understanding and forecasting shifts in the marine community, this study aims to: (I) provide the first 1-year baseline assessment of benthic assemblage composition and diversity of intertidal lateritic reefs on the Eastern Marine Ecoregion of Brazil; (ii) evaluate the role of substrate complexity and oceanic-climatic variables to benthic assemblage variability; (iii) integrate assemblage multivariate covariance factors to predict future biodiversity changes in these coastal reefs in warming scenarios for the Eastern Marine Ecoregion of Brazil. We tested the hypothesis that rugosity would have a positive influence on local benthic biodiversity, but that these effects would change temporally due to climate forcing. We expected that temperature would have a stronger effect on the variability of benthic taxa compared to the other oceanographic parameters, therefore we expected that projected warming would be a driver of assemblage composition in different climatic scenarios.

## MATERIALS & METHODS

### Study area
This study was carried out in a marine protected area in the Eastern Brazil Marine Ecoregion (Área de Proteção Costa das Algas; environmental permit by Instituto Chico Mendes # 57819-1; Fig. 1). The coastal zone is characterized by dispersed intertidal lateritic reefs with abundant macroalgal and rhodolith beds. Coastal oceanographic conditions are typically influenced by E-NE winds from the South Atlantic high-pressure system, strong internal tidal currents, and E-SE wave swells (*Pereira et al., 2005*; *Pianca, Mazzini & Siegle, 2010*). Meteorological cold fronts occur periodically and influence the vertical mixing of the water column and wave action on the coast (*Pianca, Mazzini & Siegle, 2010*). Episodic upwelling events occur mostly during spring and summer (*Pereira et al., 2005*). This is a tropical region with an average air temperature of 25 °C that has experienced significant warming trends in the last four decades (*Bernardino et al., 2015*; *Bernardino et al., 2016*).

We monitored benthic assemblages of four intertidal reefs (study sites) monthly from December 2017 to August 2018 (Fig. 1; sampling dates in Table S1). These sites are similar

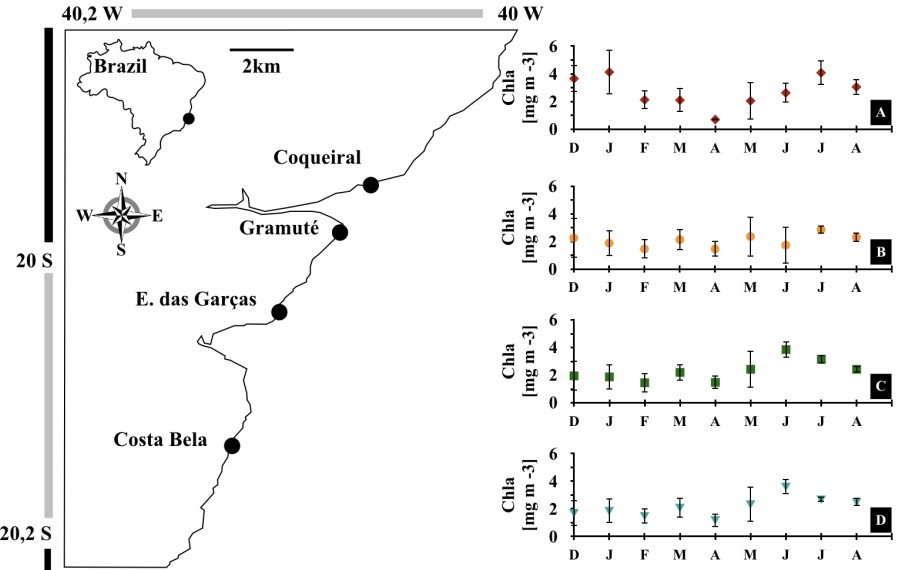

**Figure 1 Study area.** Location of the study area (APA Costa das Algas) and the four study sites in the Eastern coast of Brazil. Monthly variations in sea surface chlorophyll-a concentrations [mg m$^{-3}$] during the study (December 2017 to August 2018) at each site, Coqueiral (A), Gramuté (B), Enseada das Garças (C), and Costa Bela (D). Note: symbols and error bars represent local (site) averages per month and standard deviations, respectively.

in their presence of lateritic substrate, geographical orientation (i.e., to the East), moderate exposure to wave action, and are under similar anthropogenic pressure (i.e., located near small human settlements with no direct sewage disposal to the reef area).

## Benthic biodiversity and substrate structure

Reef benthic biodiversity (incrusting and slow mobile taxa) was estimated monthly at each site along four replicated 20-m length photo-transects using a GoPro® camera. Images were obtained by photographing from above the area delimited by a 0.5 m × 0.5 m quadrat, taking 10 photos at every 2 m along the transect. Transects were positioned parallel to the reef fringe approximately 5 m apart. Images were processed in Coral Point Count with Excel Extensions Software (CPCe) where benthic organisms (primary cover) were visually identified under 20 random points within a 100 point grid (*Carleton & Done, 1995*; *Lenth, 2001*). Quality control of the photo-processing included multiple reviewers (3) analyzing photograph subsets and in situ validation.

The substrate rugosity of the intertidal reefs was measured once at each reef by determining the linear roughness (R) through 3D modeling (*Young et al., 2017*). Carrying out rugosity measurements monthly was not logistically feasible and we assumed that reef rugosity (i.e., the substrate physical structure) was relatively stable in the temporal scale of this study. A representative intertidal area of 2 m$^2$ positioned in the area of the photo-transects in each site was photographed digitally by a series of photos which were overlaid to build a 3D model. The camera was pointed directly to the substrate with constant height and orientation (*Young et al., 2017*). The 3D models of the reefs were built

in Agisoft PhotoScan Standart Edition software through the following steps: (1) photo alignment, (2) cloud construction of points, (3) mesh, and (4) texture constructions. The site substrate roughness (R) was calculated through the virtual chain method by averaging the linear contours (linear roughness, $n = 6$) along with each 3D model using the Rhinoceros software.

## Meteo-oceanographic monitoring

Meteorological and oceanographic conditions (air and sea surface temperatures *AirT/SST*, precipitable atmosphere water *Precipitation*, significant wave height *SWH*, sea surface chlorophyll-a *Chla*, ocean salinity *Salinity*) in the study region were monitored synoptically through satellite remote sensing at the local and regional scales (4 to 30 km) (Table 1). The meteo-oceanographic data (AirT, SST, Precipitation, SWH, Salinity) were averaged for the study region (Área de Proteção Ambiental Costa das Algas, Long. 40.259 to 39.8 W, Lat. 20.3° to 19.8° S). Chla was obtained for each site (Coqueiral, Long. 40.10 W, Lat. 19.93 S; Gramuté, Long. 40.14 W, Lat. 20.02 S; E. das Garças, Long. 40.14 W, Lat. 20.06 S, Costa Bela, Long. 40.14 W, Lat. 20.10 S). The data available for the study area was then averaged monthly and included all available data for 29–30 days prior to each sampling.

## Data analysis

Temporal (monthly) differences in meteo-oceanographic conditions (AirT, SST, Precipitation, Salinity, SWH) were evaluated through a one-way balanced analysis of variance (ANOVA) (*Underwood, 1997*). For Chla, temporal (monthly) and spatial (sites) differences were tested using a two-way balanced ANOVA. A total of eight months were included in the ANOVAs, January/2017 samplings were not performed at all sites and were excluded from these analyses (Table S1). The number of replicates within a month varied according to the temporal resolution for each variable ($n = 29$ to 30 for AirT, SST, and Precipitation; $n = 672$ for SWH; $n = 4$ for Chla; $n = 3$ for Salinity). Co-variation between SST and AirT and Salinity and Precipitation were evaluated through correlation analysis (*Sokal & Rohlf, 2003*). The temporal autocorrelation in meteo-oceanographic conditions was assessed by comparing each variable versus time with correlation analyses. Variables were considered autocorrelated when they were significantly correlated with time (correlation coefficient $\neq 0$ with $p \leq 0.05$). Only Precipitation was temporally autocorrelated; this was corrected in analyses by using the differences between the averages for consecutive time periods following *Pineda & López, 2002* (Table S2). Differences in reef rugosity between sites were also evaluated through a one-way balanced ANOVA, with site as a fixed factor and the linear roughness as replicates ($n = 6$). Data were transformed ($Logx + 1$) when needed to fit the assumptions of ANOVA (normality and homogeneity of variances), verified by Shapiro–Wilk's and Cochran tests. ANOVA results were complemented by post-hoc pairwise Tukey HSD tests (*Tukey, 1949*).

Benthic assemblages were analyzed through taxa percentage cover (%) and multivariate analysis to test for spatial and temporal differences in assemblage composition and two univariate response variables: diversity (Shannon–Weiner index) and richness (the number of taxa). Biodiversity variables were calculated monthly over 10 0.5 m$^2$ quadrats per transect

Mazzuco et al. (2020), *PeerJ*, DOI 10.7717/peerj.8289

**Table 1  Meteo-oceanographic variables monitored during the study.**

| | Abbreviation | Source | Spatio-temporal resolution |
|---|---|---|---|
| Air temperature | AirT | NCEP NCAR Reanalysis (*Kalnay et al., 1996*) | grid of 2,5° latitude × longitude |
| Sea surface temperature | SST | NOAA High-resolution Blended Analysis of Daily SST (*Reynolds et al., 2007*) | a grid of 0.25° latitude × 0.25° longitude |
| Precipitable atmosphere water | Precipitation | NCEP NCAR Reanalysis (*Kalnay et al., 1996*) | grid of 2,5° latitude × longitude |
| Ocean salinity | Salinity | microwave imaging radiometer on SMOS mission's satellites (CATDS database, *Jacquette et al. (2010)*) | grid of 0.25° latitude × 0.25° longitude |
| Significant wave height | SWH | Wave Watch III | a grid of 1° latitude × 1° longitude |
| Sea surface chlorophyll-a concentration | Chla | ocean color radiometers on Modis-Acqua satellite (GIOVANNI database, *Acker & Leptoukh (2007)*) | 4 km$^2$ grid |

**Notes.**
Abreviations, sources, and spatio-temporal resolutions are listed.

for each site, resulting in a total of 4 replicates per site per month. Differences in biodiversity were assessed through permutational multivariate analysis of variance (PERMANOVA) on 9,999 permutations of residuals under a reduced model (*Anderson, 2006*; *Oksanen et al., 2018*). The PERMANOVAs compared the variability in biodiversity among sites (factor 1, fixed, with 4 levels) and monthly (factor 2, fixed, with 8 levels), with 4 replicates each. The assemblage taxa % cover was square-root transformed prior to analysis to reduce the influence of abundant and rare organisms (*Gotelli & Ellison, 2004*). PERMANOVAs were carried out using Euclidian distance for univariate analysis (diversity and richness) and the Bray-Curtis dissimilarity for assemblage composition, and significant results were complemented by post-hoc pairwise comparisons (*Anderson, 2017*).

Canonical analysis of principal coordinates (CAP) (*Anderson & Willis, 2003*) complemented by multidimensional scaling (*Anderson, 2001*; *McArdle & Anderson, 2001*; *Oksanen et al., 2018*) was performed to evaluate the association between biological and physical spatio-temporal patterns. The CAP was used to identify the environmental variable or group of variables that best explained the variation of benthic assemblage cover, and to determine the species that contributed most to the difference among samples (*Mazzuco et al., 2019*). Additionally, a canonical discriminant function analysis (DFA) was used to test spatial (i.e., between-site) differences that could be distinguished by the numerical relationship between biological and physical variables. The DFA function was built using variables with significant contributions to variability according to the CAP analysis. Substrate rugosity could not be included in the function since it is a constant value for each site. DFA results were interpreted based on the linear discriminant coefficients, which described the relationship between environmental conditions and benthic assemblage in the study region. Jackknife re-samplings were included in the analyses to test the accuracy of the classifications by DFA (*Tukey, 1949*).

Climate warming projections and their effect on benthic biodiversity were modeled with temperature scenarios ($-1\,°C$, $+1\,°C$, $+3°C$) based on current SST trends (*Muller-Karger et al., 2017*) and 20-year forecast sea surface anomaly range for the study region (*Chadwick et al., 2013*). We used a linear relationship between SST and benthic assemblage diversity indices and composition (Shannon–weiner H', species richness, and % Cover) as we had less than a year of monitoring. Climate projections were designed following: (1) description of the relationship between monthly biodiversity and SST variations through regression using current data to parameterize the models (input: monthly averages); (2) calculation of the expected monthly biodiversity based on the SST scenarios ($-1\,°C$, $+1\,°C$, $+3\,°C$ added to the current monthly SST average) and the regression algorithms. Projected changes (average monthly changes, %) were visually compared to the baseline information obtained in this study.

All statistical tests considered $\alpha = 0.05$ significance level. Graphical and analytical processing was performed in ODV (*Schlitzer, 2013*), Panoply 4.8.1 (*Schmunk, 2013*), Numbers (Apple Inc.) and R project (*R Core Team, 2018*) with packages: 'stats', 'GAD' (*Sandrini-Neto & Camargo, 2014*), 'vegan' (*Oksanen et al., 2018*), 'rich' (*Rossi, 2016*), 'outliers' (*Komsta, 2011*), and MASS (*Ripley et al., 2019*).

## RESULTS

### Meteo-oceanographic conditions

Monthly meteo-oceanographic conditions changed markedly throughout the study
(Figs. 1 and 2), with significant differences for air and sea temperatures, swell heights,
and chlorophyll-a concentrations (ANOVA AirT $F = 48.78$, SST $F = 120.91$, SWH
$F = 268.6$, Chla $F = 3.84$; $p < 0.05$, Table 2). Air and sea temperatures ranged between
22 and 28 °C and were correlated to each other ($r^2 = 0.92$, $p = 0.0003$; Table S3), with
maximum temperatures in the fall (Apr) and minima during winter (Aug) (Fig. 2, Table
S4). Precipitation and salinity averages varied from 44 to 36 and 30 to 37 ppm respectively,
and were not correlated ($df = 7$, $t = 0.64$, $r^2 = 0.23$, $p = 0.5407$). Significant wave heights
varied from 1 to 1.5 m, with higher heights at the beginning of summer (Dec) and during
winter (May-Jun-Jul-Aug), and lower wave heights during fall (Mar) (Fig. 2, Table S4).
Average chlorophyll-a concentration varied from 1 to 4.5 mg m$^{-3}$, being lower at the end
of summer (Feb) to fall (Mar-Apr-May), higher in winter (Jun-Jul-Aug), and negatively
related to sea temperatures (Fig. 2, Tables S3 and S4). Chlorophyll-a concentrations were
higher at the northern sites (Coqueiral and Gramuté; Fig. 1), with an average difference of
0.7 mg m$^{-3}$ ($F = 8.11$, $p = 0.0118$; Table 1; Table S5).

### Substrate rugosity and reef benthic cover

The 3D imaging of the intertidal reefs indicated differences in substrate rugosity among
the studied reefs ($F = 14.34$, $p < 0.001$; Table 2; Fig. 3). The northern reefs in Coqueiral
and Gramuté had higher rugosity when compared to Costa Bela and Enseada das Garças
to the South ($p < 0.001$; Table S5).

Benthic assemblages at the lateritic reefs were dominated by macroalgae (58 taxa),
with 26 Rhodophyta species, 15 Chlorophyta, and 17 Phaeophyta (Table S6). Other taxa
occurring on the reefs included cnidarians, bivalves, barnacles, hermit crabs, sea stars and
urchins, gastropods, and sponges (Table S6). The intertidal reefs were mostly covered with
macroalgae (25 to 70%) and cnidarians (Anthozoa: 12 to 26%) (Fig. 4). Some species were
sampled throughout the study at all sites, including encrusting calcareous algae, *Ulva* sp.,
*Sargassum* sp., and zoanthids (Fig. 4). Monthly trends in benthic assemblage composition
were highly variable at each site with no clear seasonal pattern but we observed an overall
higher taxa richness and diversity during fall (Figs. 3 and 4).

Benthic assemblages varied spatially in percent cover and species dominance (Figs. 3
and 4) (site $F = 87.6$ and $p = 0.01$, Table 2). Between-site variability included differences
in the live coverage (versus bare rock or sand; Fig. 4) as well as changes in the taxonomic
dominance (Fig. 3). For instance, we observed a higher Rhodophyta species richness at
Gramuté and Coqueiral, while Chlorophyta and Phaeophyta were more diverse at Gramuté
and E. das Garças (Fig. 3). Within-taxon variability was lower in Anthozoa, which was
mainly represented by *Zoanthus sociatus* and *Palythoa tuberculosa*. An exception was at
Coqueiral reefs where corals were nearly absent (Fig. 3). Overall, taxa diversity was higher
at Coqueiral ($H' = 0.94$) and Gramuté ($H' = 0.88$; $p < 0.01$; Table S7 and Fig. 4).

Benthic assemblage taxa composition also changed at temporal (monthly) scales
($F = 20.78$, $p = 0.01$; Table 2) with significant variability in the fall (Apr–May) due to

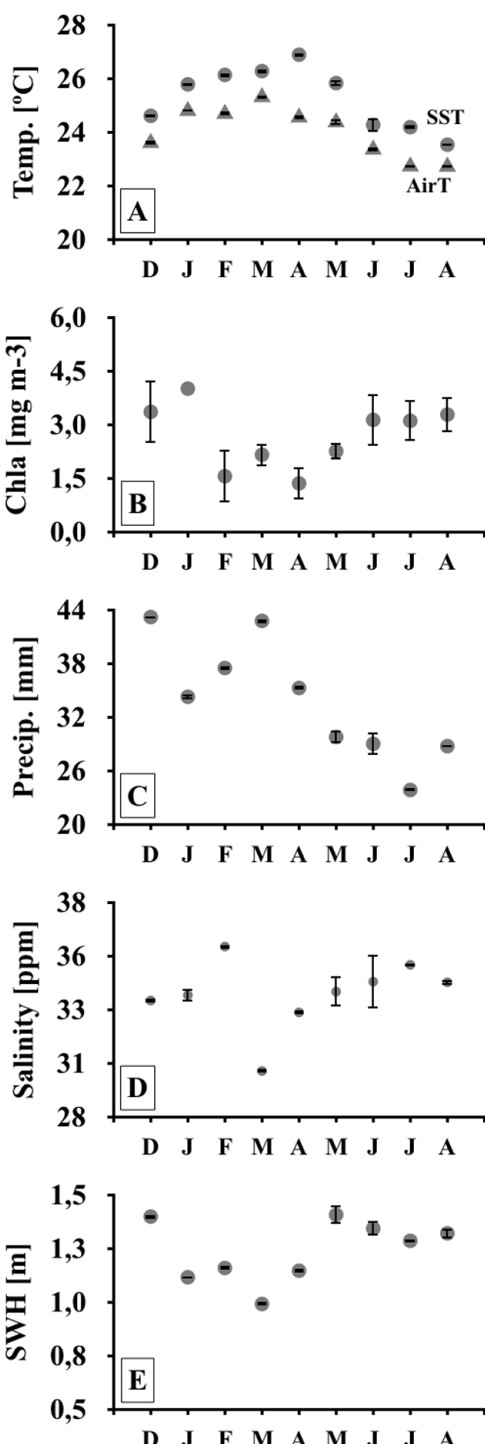

**Figure 2** **Meteo-oceanographic conditions.** Meteo-oceanographic conditions monitored during the study. Air and sea surface temperatures (Temp., AirT, SST; A, B), chlorophyll-a concentrations (Chla; C), precipitable water (Precip. D), salinity (E), and significant wave height (SWH) (F). Note: symbols and error bars represent regional (4–30 km) averages per month and standard deviations, respectively.

**Table 2  Results of ANOVAs and PERMANOVAs comparing the variability in meteo-oceanographic conditions substrate rugosity, and benthic assemblage between months and/or among sites.** Meteo-oceanographic conditions: air and sea surface temperatures AirT and SST, Precipitation, Salinity, significant wave height SWH, chlorophyll-a concentrations Chla. Benthic assemblage measures: species cover (%), diversity (Shannon–Wienner), and richness (number of taxa per quadrat per month).

| ANOVA | df | F | p | PERMANOVA | df | F | p |
|---|---|---|---|---|---|---|---|
| **Meteo-oceanographic conditions** | | | | **Benthic Assemblage** | | | |
| AirT | | | | % Cover | | | |
| Month | 8 | 48.78 | **<0.0001** | Site | 3 | 87.58 | **0.01** |
| Residual | 252 | | | Month | 7 | 20.78 | **0.01** |
| SST | | | | M*S | 21 | 8.51 | **0.01** |
| Month | 8 | 0.26 | 0.9760 | Residual | 96 | | |
| Residual | 252 | | | | | | |
| Precipitation | | | | Diversity | | | |
| Month | 8 | 25.04 | **<0.0001** | Site | 3 | 8.67 | **0.01** |
| Residual | 252 | | | Month | 7 | 7.88 | **0.01** |
| Salinity | | | | M*S | 21 | 5.14 | **0.01** |
| Month | 8 | 1.10 | 0.4072 | Residual | 96 | | |
| Residual | 18 | | | | | | |
| SWH | | | | Richness | | | |
| Month | 8 | 268.6 | **<0.0001** | Site | 3 | 11.95 | **0.01** |
| Residual | 6048 | | | Month | 7 | 26.82 | **0.01** |
| Chla | | | | M*S | 21 | 8.46 | **0.01** |
| Site | 3 | 3.84 | **0.0118** | Residual | 96 | | |
| Month | 8 | 8.11 | **<0.0001** | | | | |
| M*S | 24 | 1.86 | **0.0165** | | | | |
| Residual | 108 | | | | | | |
| Rugosity | | | | | | | |
| Site | 3 | 14.34 | **<0.0001** | | | | |
| Residual | 20 | | | | | | |

**Notes.**
F for the statistics, df for degrees of freedom. Significant results ($p < 0.05$) are in bold. Data was log-transformed Log $x + 1$ prior to the analyses.

increased macroalgal coverage ($p = 0.028$, Table S8). Taxon diversity and richness were also higher during fall ($p < 0.05$, Tables S8 and S9; Fig. 4), but monthly variations in species cover were similar across sites ($p > 0.05$, Table S9).

## Multivariate analysis

Sea surface temperatures and substrate rugosity were significantly associated with benthic assemblage composition ($p < 0.05$, Table 3, Fig. 5). The CAP ordination showed that benthic cover at Coqueiral and Gramuté were similar with higher contributions of turf, *Sargassum* sp., *P. gymnosperma*, *D. marginata*, and sediment/rock. The other two sites had higher contributions of anthozoans, which were positively related to sea surface temperature and rugosity (Fig. 5). Between-site differences could be distinguished by the set of variables pointed as significant by CAP, including sea surface temperature, benthic assemblage cover (*Z. sociatus*, *P. tuberculosa*, *Sargassum* sp., *D. marginata*, *J. rubens*, *P. gymnosperma*,

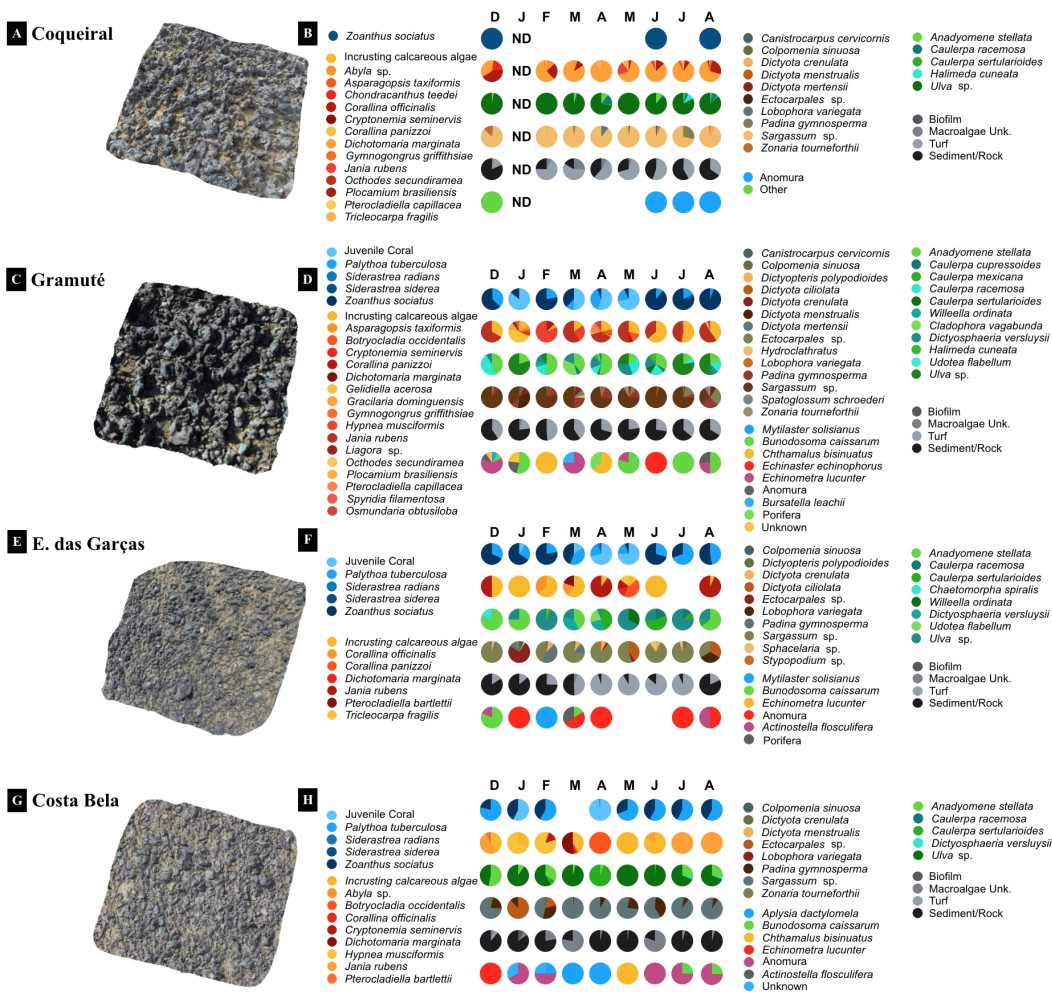

**Figure 3  3D imaging of the intertidal reefs and benthic assemblages.** 3D imaging of the intertidal reefs and taxonomic composition of the benthic assemblages at Coqueiral (A–B), Gramuté (C–D), Enseada das Garças (E–F), and Costa Bela (G–H). Note: surface contours are represented in a natural color and species diversity is highlighted with color gradients. ND stands for no data available.

*Anthozoa, turf* ), and sediment/rock cover (Table 4). The precision of classification ranged from 56% to 88%, being higher at Coqueiral and Gramuté. According to the discriminant coefficients, most variables were positively related to sea surface temperatures, with the exception of *P. tuberculosa* and *D. marginata* covers (Table 4).

The projected climatic scenarios changed the reef assemblage composition at multiple spatial scales (Fig. 6; Table S10). Changes were expected on the taxa composition (% cover from 1 to 100%); also taxa diversity and richness (0.2 to 30%). Sites with lower substrate rugosity showed similar trends of assemblage change (positive or negative) for all taxa as well as for diversity and richness. Overall, reduced regional diversity (minus 13% with +1 °C) and Anthozoa cover (minus 43% with +3 °C) were expected with increased temperature. At the local scale, forecasted higher temperatures reduced Anthozoa cover across multiple sites (minus 14% to 116%). Positive effects on macroalgae and other

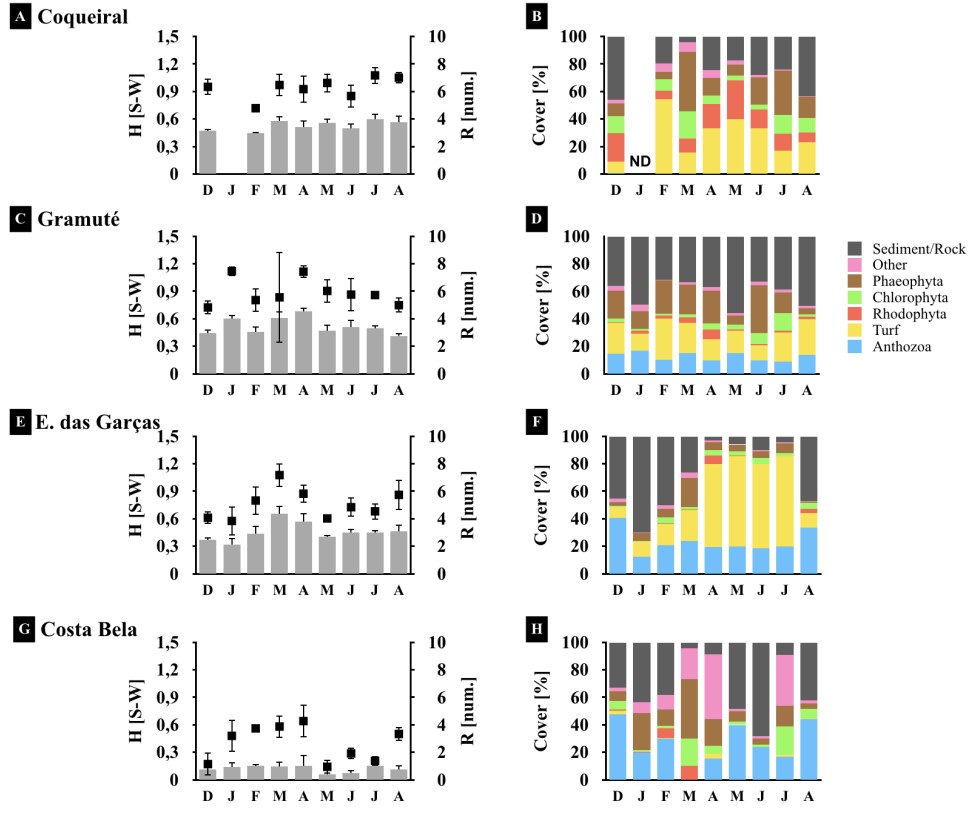

**Figure 4** **Spatio-temporal variations in the reef benthic biodiversity.** Monthly diversity (A, C, E, G) (H',
Shannon-Wiener Index S–W), richness (A, C, E, G) (number of species/taxon per quadrat per month),
and assemblage composition (B, D, F, H) (variation in % cover) at the study sites. Note: averages and
standard deviations are represented by columns and error bars respectively; color bars represent relative
cover per taxa plus sediment and rock. ND stands for no data available.

invertebrate abundance are expected with increased temperatures, both regionally and at
the sites. Macroalgae cover is expected to increase from 11 (+1 and +3 °C) to 32% (+3 °C)
regionally and up to 90% at the reefs with lower substrate rugosity (E. das Garças and
Costa Bela). Other invertebrate abundances are projected to increase from 25 to 100%
with warming. Reef assemblages (macroalgae and other invertebrates) are expected to be
negatively affected by reduced temperatures, reducing species richness and diversity as
well. In an exception, lower temperatures are expected to increase Anthozoan cover 14%
regionally and up to 100% at Coqueiral, the site with highest surface rugosity.

## DISCUSSION

Ecosystem-based indicators are the foundations for modeling and forecasting ocean
dynamics (*Miloslavich et al., 2018*). These approaches require biological data with a
reasonable taxonomic resolution and algorithms that can accurately indicate biodiversity
changes in response to environmental conditions. Our study reports a new baseline benthic
biodiversity assessment of tropical intertidal lateritic reefs on the Eastern Brazil Marine
**Table 3** **Results of canonical analyses of principal coordinates to evaluate the contribution of meteo-oceanographic conditions and benthic rugosity to month variations in the benthic assemblage composition.** Meteo-oceanographic conditions: air and sea surface temperatures AirT and SST, Precipitation, salinity, significant wave height SWH, chlorophyll-a concentrations Chla. Benthic assemblage composition measure % cover per taxa. Spearman correlation values for each environmental variable are described for in CAP axis 1–2.

| | $F = 1.72, p = 0.002$ | | | |
| --- | --- | --- | --- | --- |
| | **CAP 1 (44%)** | **CAP 2 (20%)** | **F** | **p** |
| AirT | −0.42 | 0.25 | 1.75 | 0.087 |
| SST | −0.58 | 0.37 | 1.99 | **0.052** |
| Precipitation | 0.02 | 0.04 | 1.03 | 0.393 |
| Salinity | 0.40 | −0.23 | 1.71 | 0.106 |
| SWH | 0.29 | −0.20 | 1.29 | 0.212 |
| Chla | 0.04 | −0.28 | 1.47 | 0.154 |
| Rugosity | 0.39 | 0.84 | 2.8 | **0.008** |

**Notes.**

Proportion of variability explained by CAP axes are between parentesis (), F for statistic, significant results ($p < 0.05$) are in bold.

Ecoregion that is under a decadal warming trend (*Bernardino et al., 2015*). The hypothesis that the composition and diversity of intertidal benthic assemblages would be associated with substrate rugosity and seasonal effects was validated, supporting the long-term monitoring of essential ocean variables on the South Atlantic (*Muller-Karger et al., 2017*; *Muelbert et al., 2019*).

Coastal intertidal reef benthic diversity was temporally correlated with sea surface temperatures. Variations of sea temperature are being increasingly recognized to change temporal patterns of coastal biodiversity at both seasonal and interannual time scales (*Poloczanska et al., 2013*; *Poloczanska et al., 2016*). Our findings support that tropical marine intertidal reef ecosystems may be extremely vulnerable to warming given the influence on benthic assemblage composition, especially on habitat-forming species such as anthozoans and macroalgae. Anthozoans should be the most negatively affected by higher temperatures, the major cause of bleaching, pathogen spread, and coral reef declines (*Blackwood et al., 2017*; *Sully et al., 2019*). On the other hand, years with negative temperature anomaly, observed within the warming trend, could benefit macroalgae and other invertebrates, such as sea urchins and decapods. Future benthic assemblage composition and states will depend on community feedbacks (*Steneck & Dethier, 1994*; *Duffy, 2002*; *Deáth et al., 2012*; *Lemieux & Cusson, 2014*; *Andrades et al., 2019*). Although temporal variability in benthic assemblage composition may be a matter of seasonality and turnover (*Hartnoll & Hawkins, 1980*; *Dye, 1998*; *Okuda et al., 2004*), warming temperatures could negatively affect macroalgae beds and lead to decreased reef productivity and community shifts in the long term (*Steneck & Dethier, 1994*; *Sorte et al., 2017*). Temperature effects were also observed to decrease larval diversity on the Eastern Brazil Marine Ecoregion (*Mazzuco et al., 2019*) and suggest that reduced overall benthic diversity on intertidal reefs may be associated to lower physiological fitness of both adult and larval life stages.

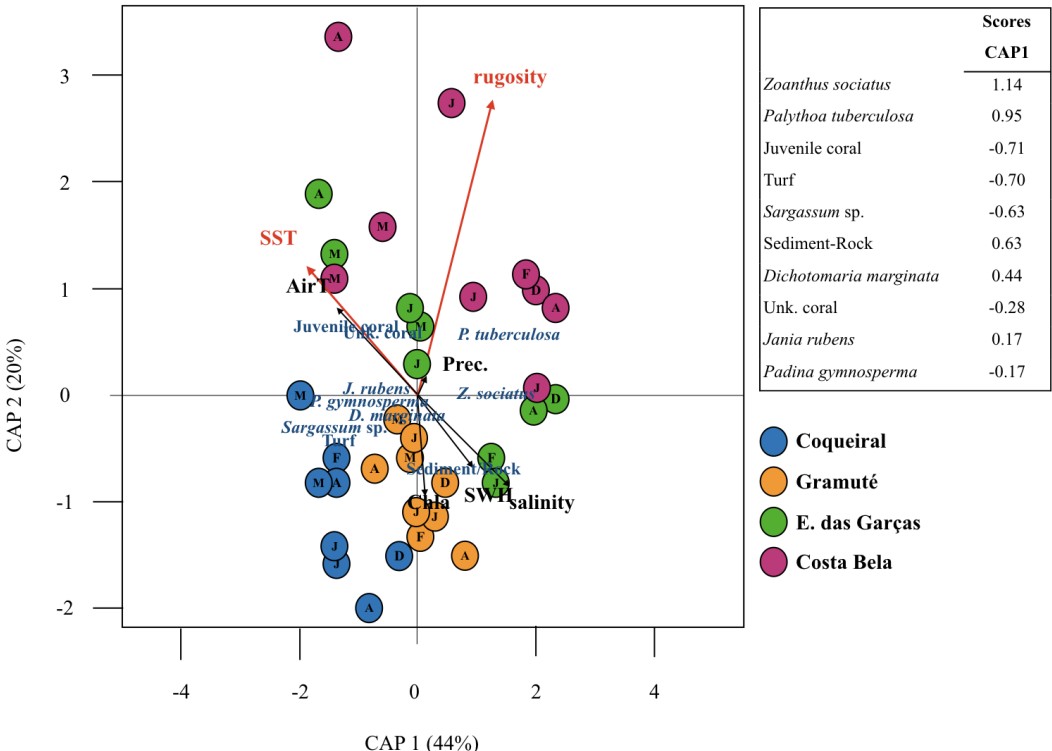

**Figure 5** **Canonical analyses of principal coordinates.** Canonical analyses of principal coordinates (CAP) indicating differences in the benthic assemblage composition (% cover per taxa) at the study sites (Coqueiral, Gramuté, E. das Garças, and Costa Bela) and the contribution of meteo-oceanographic conditions (air and sea surface temperatures AirT/SST, chlorophyll-a concentration Chla, precipitable water Precip, salinity, and significant wave height SWH) and substrate rugosity. Vectors are based on Spearman correlation values $> 0.5$ ($p < 0.5$) for environmental variables and scores for taxa. The proportion of data explained by axis 1 and 2 are in parenthesis.

Spatial patterns of benthic assemblages were driven by differences in the coverage of macroalgae and anthozoans on intertidal reefs. The outcomes of macroalgae-anthozoa shifts to biodiversity and productivity vary across ecosystems (*Norström et al., 2009*; *Chemello, Vizzini & Mazzola, 2018*). Substrate rugosity is a major driver of benthic biodiversity on rocky shores (*Guichard, Bourget & Robert, 2001*; *Cruz-Motta et al., 2010*; *Bloch & Klingbeil, 2016*) and coral reefs (*Zawada, Piniak & Hearn, 2010*). We confirmed previous interactions of substrate rugosity and benthic species richness, diversity, and composition for intertidal lateritic reefs and their strong seasonal association with sea temperatures. Besides, regional differences in diversity were positively related to substrate rugosity and corroborate assertations of higher habitat availability at more complex reefs. These patterns indicate that not only should substrate rugosity to be considered and reported when analyzing coastal reef benthic assemblages, but also that sampling that occurred across sites of variable rugosity could result in misleading biogeographic patterns.

The use of indicator taxa to distinguish benthic biodiversity allowed differentiation of spatial and temporal scales along these intertidal reefs. According to our findings, merging

**Table 4 Results of canonical discriminant function analysis to assess the between-site differences according to meteo-oceanographic conditions and benthic assemblage composition.** Meteo-oceanographic conditions: sea surface temperatures SST. Benthic assemblage composition measure: % cover per taxa. Linear discriminant coefficients LD1 and results of Jackknife re-samplings to test the accuracy of the classification for each site.

|                | LD1    | Site          | Accuracy |
|----------------|--------|---------------|----------|
| SST            | 0.32   | Coqueiral     | 88%      |
| *Z. sociatus*  | 0.17   | Gramuté       | 78%      |
| *P. tuberculosa* | −0.02 | E. das Garças | 67%      |
| Juvenile coral | 0.01   | Costa Bela    | 56%      |
| Turf           | 0.06   |               |          |
| *Sargassum* sp. | 0.08  |               |          |
| Sediment-Rock  | 0.04   |               |          |
| *D. marginata* | −0.31  |               |          |
| Unk. coral     | 0.09   |               |          |
| *J. rubens*    | 0.34   |               |          |
| *P. gymnosperma* | 0.06 |               |          |

a specific set of in situ data (benthic cover of dominant taxa; i.e., *Zoanthus sociatus*, *Paythoa tuberculosa*, *Padina gymnosperma*, and others) and satellite remote sensing products (sea surface temperature) would allow meaningful long-term assessments of benthic biodiversity in tropical reefs at large scales, improving our capacity to manage these coastal ecosystems. Expanding the monitoring zone and frequency is a current challenge in this coast with the least amount of long-term ecological research sites. This framework simplifies reef monitoring protocols for the intertidal zone and may likely help management actions, for instance with early detection of biodiversity change. Our benthic assemblage database allows species tracking, and forecasting simulations for tropical lateritic reef communities.

Generating high-resolution and long-term biodiversity information across marine ecosystems is urgent when dealing with environmental and climate emergencies (*Canonico et al., 2019*). This study not only provides a baseline open-access ecological database for the intertidal reefs in the Eastern Brazil Marine Ecoregion but also is the first attempt to project benthic biodiversity outcomes of warming for this region. The climate projections highlight a potential overall decrease in regional diversity of benthic assemblages with warming, corroborating global assessments of marine biodiversity loss (*Scheffer et al., 2001*) and the vulnerability of tropical reefs to climate change (*IPCC, in press*). If future community shifts in benthic diversity of lateritic reefs are confirmed, we should expect alterations in ecosystem trophic structure and negative impacts on ecosystem services at large scales.

## CONCLUSIONS

Our results revealed a distinct benthic assemblage on intertidal lateritic reefs on the Eastern Brazil Marine Ecoregion, showing that substrate rugosity and seasonal changes in temperature are key to taxa richness, diversity and species composition. Spatial patterns of assemblage structure were either dominated by macroalgae on more complex reefs, or

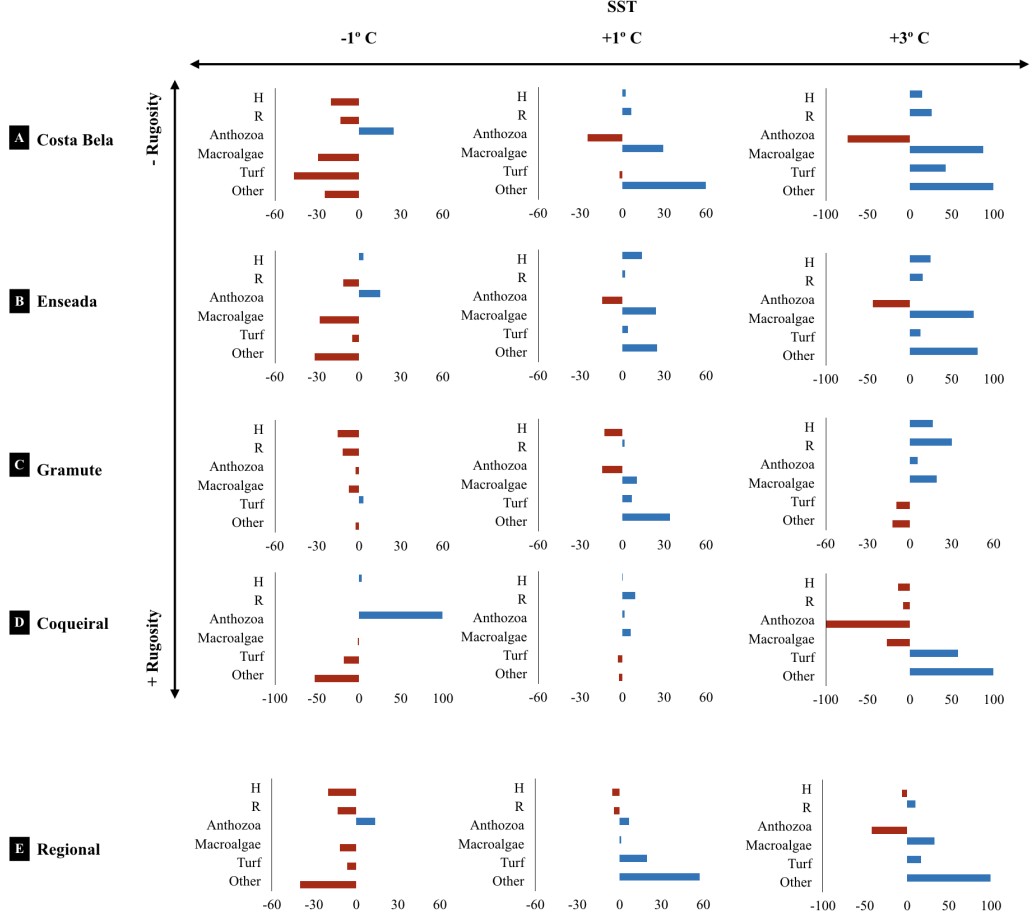

**Figure 6  Climate projected changes in benthic reef biodiversity.** Projected regional and site (Costa Bela A, Enseada das Garças B, Gramuté C, Coqueiral D) changes (%) in the benthic biodiversity (H' Diversity, Shannon-Wiener Index S-W, R' Richness number of species/taxon per quadrat per month, assemblage composition, variation in % Cover) with warming scenarios (sea surface temperature SST +1° C, +3° C, −1° C. Note: sites are organized in relation to higher (+) and lower (−) substrate rugosity.

Anthozoa, in reefs with lower rugosity. The intimate association of temperature and taxa compostion suggest that predicted warming in the Eastern Brazil Marine Ecoregion will have a major role in the loss of species in intertidal reefs on the tropics, with yet unknown consequences to ecosystem process. Our work further illustrates the utility of monitoring tools based on in situ data and satellite remote sensing products, and support long-term global efforts to improve ecosystem management.

## ACKNOWLEDGEMENTS

We thank all the co-workers from Grupo de Ecologia Bêntica for the assistance during field samplings and discussions. We would like to thank Lorena Monteiro for assisting in photo-processing. We are also grateful to the Federal University of Espírito Santo and Instituto Chico Mendes for their support. This article is the #008 contribution of The PELD

(Long-term Program of Ecological Research) Coastal Habitats of Espírito Santo as part of the Brazilian LTER program.

### Funding

This work was supported by Coordenação de Aperfeiçoamento de Pessoal de Nível Superior, Conselho Nacional de Desenvolvimento Científico e Tecnológico, and Fundação de Amparo a Pesquisa do Estado do Espírito Santo (fellowships and project grants 88887.185758/2018-00, 88887.137932/2017-00, 371008/2017-4, 441243/2016-9, 79054684/17). The funders had no role in study design, data collection and analysis, decision to publish, or preparation of the manuscript.

### Grant Disclosures

The following grant information was disclosed by the authors:
Coordenação de Aperfeiçoamento de Pessoal de Nível Superior.
Conselho Nacional de Desenvolvimento Científico e Tecnológico.
Fundação de Amparo a Pesquisa do Estado do Espírito Santo: 88887.185758/2018-00, 88887.137932/2017-00, 371008/2017-4, 441243/2016-9, 79054684/17.

### Competing Interests

The authors declare there are no competing interests.

### Author Contributions

- Ana Carolina de A. Mazzuco conceived and designed the experiments, performed the experiments, analyzed the data, prepared figures and/or tables, authored or reviewed drafts of the paper, and approved the final draft.
- Patricia Sarcinelli Stelzer conceived and designed the experiments, performed the experiments, analyzed the data, authored or reviewed drafts of the paper, and approved the final draft.
- Angelo F. Bernardino conceived and designed the experiments, analyzed the data, authored or reviewed drafts of the paper, and approved the final draft.

### Field Study Permissions

The following information was supplied relating to field study approvals (i.e., approving body and any reference numbers):
Environmental permit #57819-1 was approved by Instituto Chico Mendes ICMBio.

### Data Availability

The dataset in an online ITP with automatic output to OBIS, which will enable other scientist to conduct further studies. The data is also formatted in MBON Pole to Pole standards, where we are developing a user-friendly dashboard for decision-making and environmental management.

Link to the repository:

http://ipt.iobis.org/mbon/.

OBIS report on the dataset:

http://ipt.iobis.org/mbon/resource?r=brazil_reef_biodiversity_ilter_coastal_habitats_es_gramute.

http://ipt.iobis.org/mbon/resource?r=brazil_reef_biodiversity_ilter_coastal_habitats_es.

http://ipt.iobis.org/mbon/resource?r=brazil_reef_biodiversity_ilter_coastal_habitats_es_costabela.

http://ipt.iobis.org/mbon/resource?r=brazil_reef_biodiversity_ilter_coastal_habitats_es_enseada.

This data is also available as Supplementary Files.

## Supplemental Information

Supplemental information for this article can be found online at http://dx.doi.org/10.7717/peerj.8289#supplemental-information.

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
