# Peer review of "Substrate rugosity and temperature matters: patterns of benthic diversity at tropical intertidal reefs in the SW Atlantic"

_PeerJ, doi:10.7717/peerj.8289_

## Round 0.1 · original submission · Major Revisions

Dear authors, there are some technical and conceptual issues with your manuscript. First of all, the reviewers found that the study lacks novelty, and with the data collected you could have addressed a new problem. You have collected a huge amount of data and it should permit you to address several exiting ecological problems. You probably would like to review the existing literature on the topic, and it will give you some ideas for comparison and new directions. You can also present your data as an online database with a user-friendly interface, that will enable other scientists to conduct the analysis. In this case, you do not need to find a novel problem, you will need to present a case how to use the database.

Reviewer 1 ·

Basic reporting

Overall this manuscript suffers from some lack of clarity in writing. However, the biggest issue with this manuscript is that I am not convinced the findings are novel in any way. The fact that sea surface temperatures and rugosity impact benthic biodiversity and community structure has been demonstrated for many ecosystems from the intertidal to the deep sea. The data appear robust and the study design is sound. I simply encourage the authors to find a more compelling question to answer. For example, can their information be used to establish predictions of baseline community structure, diversity, or richness predictions based on rugosity and SST patterns (or extremes, or anything else) that managers could then use to determine if another stressor is impacting the community? Why did the authors first want to test if lateritic reefs were structured by different abiotic factors- is there a reason they thought they might be?

Experimental design

The study design seems relatively sound. However, the spatial scale over which the explanatory variables were averaged is unclear. Also, the unit of replication at each site is unclear; at one point the authors state that each 0.5 m2 quadrat was used as a replicate but later in the same paragraph they say each transect was used as a replicate. Either way, but particularly if the individual quadrats were used as the unit of replication, the data from within a site at each sampling event should be tested for spatial autocorrelation. Finally, I wonder why the authors chose to assess rugosity in one 2 m2 patch instead of over each transect or at each quadrat. Finally, the analysis presented in the methods does not perfectly match the response variables reported in the results. This should be addressed.

Validity of the findings

With the caveats of methodological details highlighted in the previous section I do not have any reason to doubt the validity of the findings. I do doubt their novelty and impact on the field. The discussion of the paper focusses mostly on what is already known about rugosity and SST shaping benthic communities and does not do a good job of highlighting what this paper contributes to this body of knowledge. The authors claim, in the first paragraph of the discussion, to have developed ecosystem-based indicators for this system. They did not do this. An ecosystem based indicator is a value that can either be measured or calculated from a set of measurements in the ecosystem that, when tracked over time, tells you information about the health or state of the ecosystem. The authors did not develop a new ecosystem-level indicator nor did they, as the work is currently presented, assess the validity of a proposed indicator.

Additional comments

The authors have gathered a large amount of, what appears to be high quality data. I encourage them to rethink their approach and use this data to address a question that is more interesting to a larger audience.

·

Basic reporting

1. Consider results of GUICHARD, F.; BOURGET, E.; ROBERT, J. L. Scaling the influence of topography heterogeneity on intertidal benthic communities: alternate trajectories mediated by hidrodinamics and shading. Marine Ecology Progress Series, 217, 27-41. 2001.

2. Figure 3 have some too small letters.

Experimental design

1. The authors sad that “sites are similar in their presence of lateritic substrate, easiness of access, geographical orientation, exposure to wave action, geomorphological features, and anthropogenic pressure”. Provide some data or reference on that.
2. Although the studied area is in a marine protected area, there are populated areas that could disturb the marine community under study. The authors should make clear whether the reefs are really comparable or whether different degrees of impact between reefs could cause some of the differences found.

3. How the transect were positioned on each reef? Randomly?
4. “A representative intertidal area of 2m2 in each site was photographed” – how these representative area were chosen?

Validity of the findings

no comment

Additional comments

The study brings relevant information on marine biodiversity at tropical reefs in the South Atlantic and its relationship with reef rugosity and sea surface temperatures. Some minor questions are pointed out.

---

## Round 0.2 · Major Revisions

At this stage it is important to re-organize the manuscript. The reviewer has provided an annotated manuscript with multiple suggestions. I urge you to implement the suggestions. It would be good to explain the idea and motivation behind the study, it will help the readers to appreciated the manuscript. Some of the figures and tables can be moved to Supplement. If you streamline the paper focusing on supporting the scientific claims that you make, the paper will be much better received.

Reviewer 1 ·

Basic reporting

This paper still needs to be streamlined and better supported. I still feel the introduction does not set up the study well- by the end of it I still have no idea why the authors did the study. There are a couple instances of ambiguity in the language used and many instances where the language is general and vague where specifics would be helpful. I have pointed these out in the detailed comments I have provided with this review. Additionally, in the discussion the authors make several claims of causation where their data only support correlation. Again, these are pointed out in the comments I have provided. Finally, the authors only state two hypotheses, but there are 6 tables and 6 figures. This seems excessive. Particularly as some data is presented in both table and figure form.

Experimental design

Experimental design is sufficient.

Validity of the findings

The authors do not do a good job of putting their specific findings in the context of what is known. In the discussion there are very few instances where specifics of the studies findings are mentioned. When they are statements implying causation are used where only correlation is supported by the study.

Annotated reviews are not available for download in order to protect the identity of reviewers who chose to remain anonymous.

---

## Round 0.3 · Minor Revisions

Could you please work on the introduction and abstract to make sure the paper is clearly written. The reviewer has provided some useful suggestions in the annotated manuscript.

Reviewer 1 ·

Basic reporting

Overall the paper is greatly improved. There are still some issues with clarity of writing- largely in the introduction.

Experimental design

It is fine.

Validity of the findings

Much improved.

Annotated reviews are not available for download in order to protect the identity of reviewers who chose to remain anonymous.

---

## Round 0.4 · accepted · Accept

Thank you for addressing all the raised concerns. I would like to recommend the paper for acceptance.